# Activation of the JNK/COX-2/HIF-1α axis promotes M1 macrophage via glycolytic shift in HIV-1 infection

Junhan Zhang*, Zongxiang Yuan*, Xuanrong Li, Fengyi Wang, Xueqin Wei, Yiwen Kang, Chuye Mo, Junjun Jiang, Hao Liang [iD], Li Ye [iD]

Chronic inflammation is recognized as a major risk factor for the severity of HIV infection. Whether metabolism reprogramming of macrophages caused by HIV-1 is related to chronic inflammatory activation, especially M1 polarization of macrophages, is inconclusive. Here, we show that HIV-1 infection induces M1 polarization and enhanced glycolysis in macrophages. Blockade of glycolysis inhibits M1 polarization of macrophages, indicating that HIV-1–induced M1 polarization is supported by enhanced glycolysis. Moreover, we find that this immunometabolic adaptation is dependent on hypoxia-inducible factor 1α (HIF-1α), a strong inducer of glycolysis. HIF-1α–target genes, including HK2, PDK1, and LDHA, are also involved in this process. Further research discovers that COX-2 regulates HIF-1α–dependent glycolysis. However, the elevated expression of COX-2, enhanced glycolysis, and M1 polarization of macrophages could be reversed by inactivation of JNK in the context of HIV-1 infection. Our study mechanistically elucidates that the JNK/COX-2/HIF-1α axis is activated to strengthen glycolysis, thereby promoting M1 polarization in macrophages in HIV-1 infection, providing a new idea for resolving chronic inflammation in clinical AIDS patients.

## Introduction

Chronic inflammation is a hallmark of HIV-1 infection, and it persists even when combination antiretroviral therapy is effective (Deeks et al, 2013). This persistent inflammation is now widely recognized as a major risk factor for the severity of HIV infection, which can exacerbate tissue damage and lead to HIV-related non-communicable complications such as cardiovascular diseases, liver and kidney diseases, malignant tumors, and potentially fatal outcomes (Deeks et al, 2013). Although some anti-inflammatory drugs have been applied clinically to address chronic inflammation in HIV-1–infected patients, the therapeutic effect is not satisfactory (O'Brien et al, 2017; Utay & Overton, 2020). Therefore, there is an urgent need to better understand the mechanisms driving chronic inflammation in HIV-1 infection and to discover new therapeutic intervention targets.

In the past few years, the role of cellular metabolism in the fate and activity of immune cells has been revealed, including its influence on the outcome of infectious diseases. Increasing evidence indicates that immunometabolism plays an important role in HIV-1 pathogenesis. It is believed that HIV-1 infection is favored in cells with high levels of metabolic activities (Sáez-Cirión et al, 2021). Some studies have shown that HIV-1–infected CD4+ T cells are characterized by increased glycolysis and glutaminolysis, which support HIV-1 infection (Hegedus et al, 2014, 2017). On the contrary, different metabolic characteristics of HIV-specific CD8+ T cells from non-controllers and from HIV controllers (HICs) have been revealed. Based on this, researchers proposed to reprogram the metabolic profile of CD8+ T cells as a strategy for HIV-1 cure (Angin et al, 2019; Perdomo-Celis et al, 2022). As in T cells, altered metabolic profiles have also been observed in HIV-1–infected macrophages (Datta et al, 2016). Chronic inflammation in HIV-1 infection is closely related to the activity of immune cells, particularly macrophages.

Macrophages are not only the target cells of HIV-1 infection, but also the key regulators of chronic immune activation in HIV/AIDS patients, especially in the advanced stage of HIV-1 infection. Because of their longevity, macrophages play a more prominent immunomodulatory role compared with the depleted T cells (Cashin et al, 2011). In response to microenvironmental changes, macrophages undergo polarization. This polarization process, resulting in classically activated (M1) and alternatively activated (M2) macrophages, is critical in mediating effective immune responses against pathogen invasion. M1 macrophages participate in pro-inflammatory response after being activated by Th1 cytokines, pro-inflammatory cytokines, and chemokines, whereas M2 macrophages promote anti-inflammatory response and tissue damage repair (Cassol et al, 2009). Some pathogens can use these activation pathways to facilitate dissemination and pathogenesis. As for HIV-1, imbalanced polarization of macrophages is a key mechanism for chronic inflammation and immune activation in HIV/AIDS patients (Teer et al, 2021).

Guangxi Key Laboratory of AIDS Prevention and Treatment, School of Public Health, Guangxi Medical University, Nanning, China

Correspondence: jiangjunjun@gxmu.edu.cn; lianghao@gxmu.edu.cn; yeli@gxmu.edu.cn
*Junhan Zhang and Zongxiang Yuan contributed equally to this work

Along with polarization, macrophages initiate adaptive reprogramming of metabolism in response to antigen stimuli through rapid changes in key genes and enzymes in metabolic pathways (O'Neill et al, 2016; Saha et al, 2017). Some studies have demonstrated that macrophage polarization is tightly bound up with the alteration in glucose metabolism. For example, LPS-induced M1 macrophages use glycolysis as the main energy supplier because of the high demand for rapid ATP synthesis. This phenomenon is similar to the well-known Warburg effect or aerobic glycolysis (Palsson-McDermott et al, 2015). In contrast, in M2 macrophages, oxidative phosphorylation (OXPHOS) can better meet the continuous intracellular energy supply required by macrophages for tissue repair and wound healing (Viola et al, 2019; Wang et al, 2018). An earlier study showed that HIV infection can induce unique metabolic characteristics in macrophages, including lipid accumulation and reduced mitochondrial ATP production (Castellano et al, 2019). In addition, metabolic reprogramming, as a result of HIV-1 infection, is conducive to the regulation of the cellular immune function (Sáez-Cirión et al, 2021). As for macrophages, the inherent plasticity provides a basis for macrophage-centered therapeutic approaches (Kim et al, 2021). However, there is limited information on the precise mechanism(s). Therefore, it is very attractive to obtain in-depth insights into the effects of metabolic reprogramming on macrophage polarization in HIV-1 infection.

In this study, we investigated the polarization state of macrophages upon HIV-1 infection and found that the M1 subtype was induced along with the secretion of various pro-inflammatory cytokines. In the metabolic analysis, we found that glucose metabolic reprogramming occurred in HIV-1 infection, with enhanced glycolysis and reduced TCA and OXPHOS. Mechanistically, we found that HIV-1 activates JNK/COX-2/hypoxia-inducible factor 1$\alpha$ (HIF-1$\alpha$) signaling to alter glucose metabolism, thereby facilitating M1 polarization in macrophages. Our results reveal new understandings of immunometabolism in macrophages in response to HIV-1 infection and provide novel strategies for clinical intervention of chronic inflammation in HIV/AIDS patients.

# Results

### HIV-1 infection induces M1 polarization of macrophages

THP-1 cells or monocytes from healthy donors were respectively differentiated into macrophages in vitro, for the following infection experiments. The morphology of HIV-1–infected monocyte-derived macrophages (MDMs) under an inverted optical microscope showed that macrophages gradually developed an irregular shape and had prolonged long pseudopodia post–HIV-1 infection (Fig 1A), similar to LPS-stimulated M1 macrophages (Fig S1A). In addition, the mRNA expression of M1 markers, including TNF-$\alpha$, IL-1$\beta$, and IL-6, was significantly elevated in HIV-1–infected cells compared with the control group (Figs 1B and S1B). Although the M2 markers CD163, CD206, and IL-10 increased simultaneously, the magnitude of the increase was much lower than that of the M1 markers (Fig 1B). At the protein level, we observed these M1 inflammatory factors (TNF-$\alpha$, IL-

1$\beta$, and IL-6) sustained the release to the supernatant during HIV-1 infection (Fig 1C). Furthermore, flow cytometric analyses revealed an increased proportion of CD68$^+$CD86$^+$ M1 macrophages after HIV-1 infection (33.29% versus 67.60%), whereas the proportion of CD68$^+$CD163$^+$ M2 macrophages was just slightly increased (7.18% versus 12.33%), and the proportion of CD68$^+$CD163$^+$ and CD68$^+$IL-10$^+$ M2 macrophages was decreased after HIV-1 infection (22.61% versus 11.56%; 2.18% versus 7.80%, respectively) (Fig 1D). Taken together, the above evidence confirms that HIV-1 infection induces M1 polarization of macrophages.

### Changes in glucose metabolism in HIV-1–infected macrophages

Previous studies have shown that the metabolic reprogramming is involved in polarization of macrophages (Russell et al, 2019). Changes in the ATP/ADP ratio can indicate the primary energy metabolic pathway in the cell (Maldonado & Lemasters, 2014; Tan et al, 2022). Therefore, we used the ATP/ADP ratio to determine the metabolic state of HIV-1–infected macrophages. As a result, a lower ATP/ADP ratio was observed under HIV-1 infection, which indicated the cell metabolism is predominantly glycolytic (Fig 2A, left panel). Besides that, we measured the relative ATP production from glycolysis and OXPHOS in the context of HIV-1 infection in macrophages via inhibition of two ATP production pathways. Inhibitor 2-DG (10 mM), targeting hexokinase II (HK2), was used to block glycolysis (Pajak et al, 2019), and inhibitor oligomycin (40 $\mu$M) was used to block mitochondrial ATP production (Nieminen et al, 1994). Inhibition of glycolysis with 2-DG resulted in a greater ATP reduction in HIV-1–infected MDMs compared with control cells (HIV-1: 51.2% versus Control: 40.6%), whereas oligomycin inhibition of mitochondrial ATP production had a greater effect on reduction of ATP in uninfected control cells (Fig 2A, right panel), revealing that HIV-1 facilitates the use of glycolysis for energy supply.

To verify the changes in glucose metabolism pathways by HIV-1 infection, glucose uptake was detected. The results showed a significant increase in the proportion of glucose uptake in HIV-1–infected cells compared with control cells (HIV-1: 23.44% versus Control: 1.66%) (Fig 2B). As the final metabolic product of glycolysis, lactate accumulation means the increase in glycolysis. In the HIV-1–infected group, the level of lactate in the supernatant was about 1.7 times that in the control group (Fig 2C), and the results of the extracellular acidification assay yielded identical results (Fig 2D). More significantly, the latter showed a time-dependent effect. Meanwhile, we performed the oxygen consumption assay and the JC-1 mitochondrial membrane potential (MMP) assay to estimate the mitochondrial activity during HIV-1 infection. HIV infection led to a significant decrease in the oxygen consumption rate (OCR) and a 40% reduction in MMP, indicating impaired mitochondrial activity (Fig 2E and F). By calculating glycolytic indexes (lactate production × glucose uptake rate/OCR) (Valentín-Guillama et al, 2018), comprehensively, we revealed that glycolysis was enhanced in HIV-1–infected MDMs (Fig 2G).

In addition, we detected a few key genes for TCA and OXPHOS in HIV-1–infected MDMs. Intriguingly, the expression of all these genes was down-regulated. Among them, isocitrate dehydrogenase (IDH) was down-regulated by about two times ($P < 0.01$), and malate dehydrogenase was down-regulated by about 1.6 times ($P < 0.05$) (Fig 2H). IDH and malate dehydrogenase are both important rate-

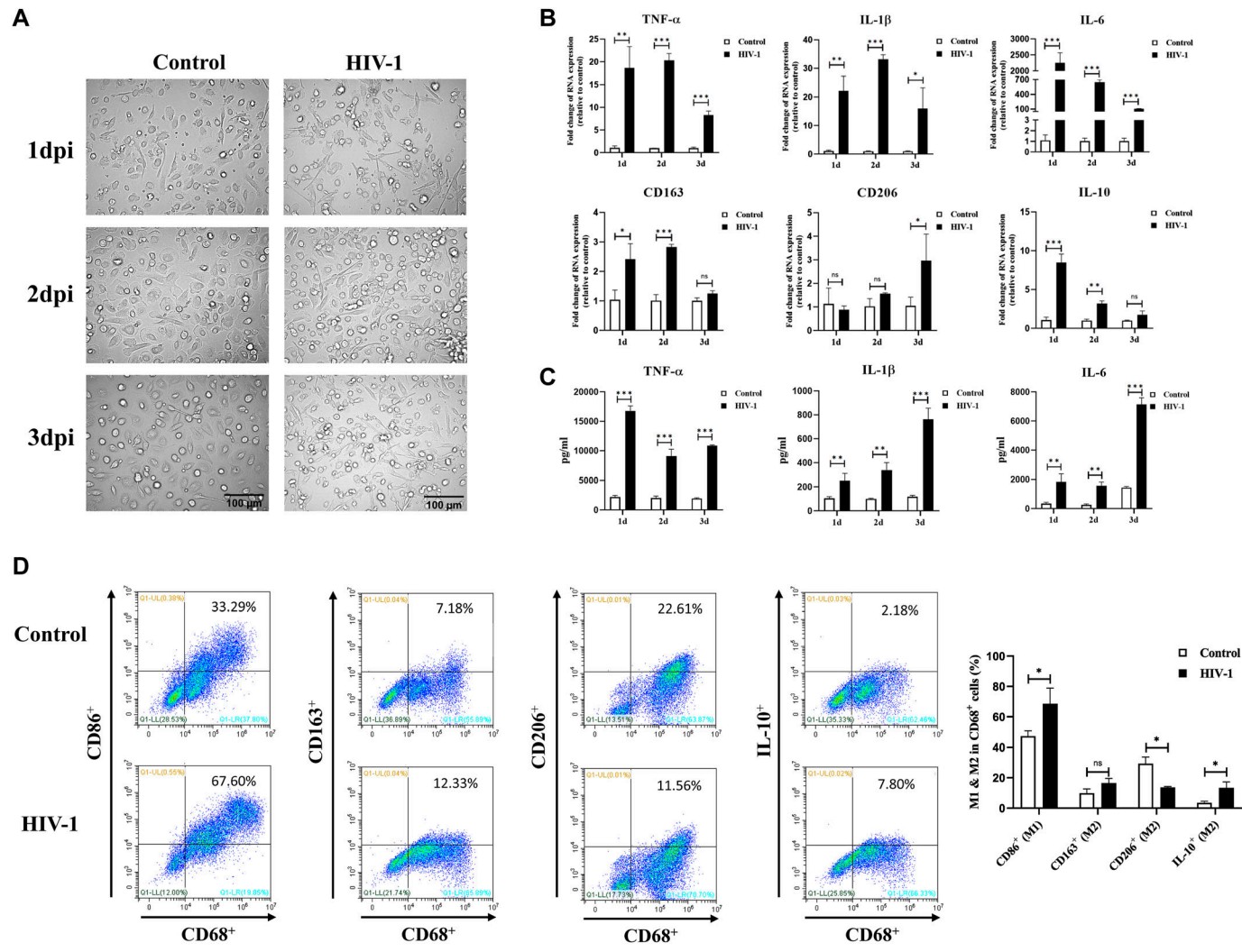

**Figure 1. HIV-1 infection induces M1 polarization of macrophages.**
**(A)** Morphologic features of monocyte-derived macrophages (MDMs) after HIV-1 infection for 1–3 d. Scale bar, 100 μm. **(B, C)** Changes in mRNA (B) and protein expressions (C) of M1 and M2 polarization markers in MDMs after HIV-1 infection for 1–3 d. (Statistical analysis was performed using a *t* test, *P < 0.05, **P < 0.01, and ***P < 0.001.) **(D)** Representative results of expressions of M1 and M2 polarization markers in MDMs detected by flow cytometry. (Statistical analysis was performed using a *t* test, *P < 0.05).

limiting enzymes in the TCA cycle. IDH catalyzes the synthesis of α-ketoglutarate, which is the intermediate product connecting carbon metabolism and nitrogen metabolism in the body.

The above results suggest that HIV-1 infection may alter the glucose metabolism of macrophages by promoting glycolysis and inhibiting TCA and OXPHOS.

### Suppression of glycolysis blocks M1 polarization of macrophages in HIV-1 infection

In multiple reports, M1 macrophages are characterized by increased glycolytic activity, whereas M2 macrophages are characterized by increased fatty acid oxidation (FAO), glutaminolysis, and mito-chondrial respiration (Viola et al, 2019). To investigate the impact of metabolic reprogramming in HIV-1–induced M1 polarization, we

used 2-DG (targeting HK2) and heptelidic acid (targeting GAPDH) to block glycolysis. As shown in Figs 2I and S2A, both 2-DG and heptelidic acid treatment effectively reduced the elevated lactate release by HIV-1 infection. Under the condition of glycolysis inhi-bition, we observed that the expression of M1 markers TNF-α, IL-1β, and IL-6 was also reversed (Figs 2J and S2B–D), indicating that blocking glycolysis inhibits M1 polarization of macrophages driven by HIV-1 infection. In contrast, treatment of oligomycin promoted the release of pro-inflammatory cytokines, as evidenced in Fig S2.

### HIV-1 infection leads to immunometabolic reprogramming in macrophages via activation of HIF-1α

To further study the molecular mechanism of immunometabolic reprogramming by HIV-1 infection, we investigated the expression

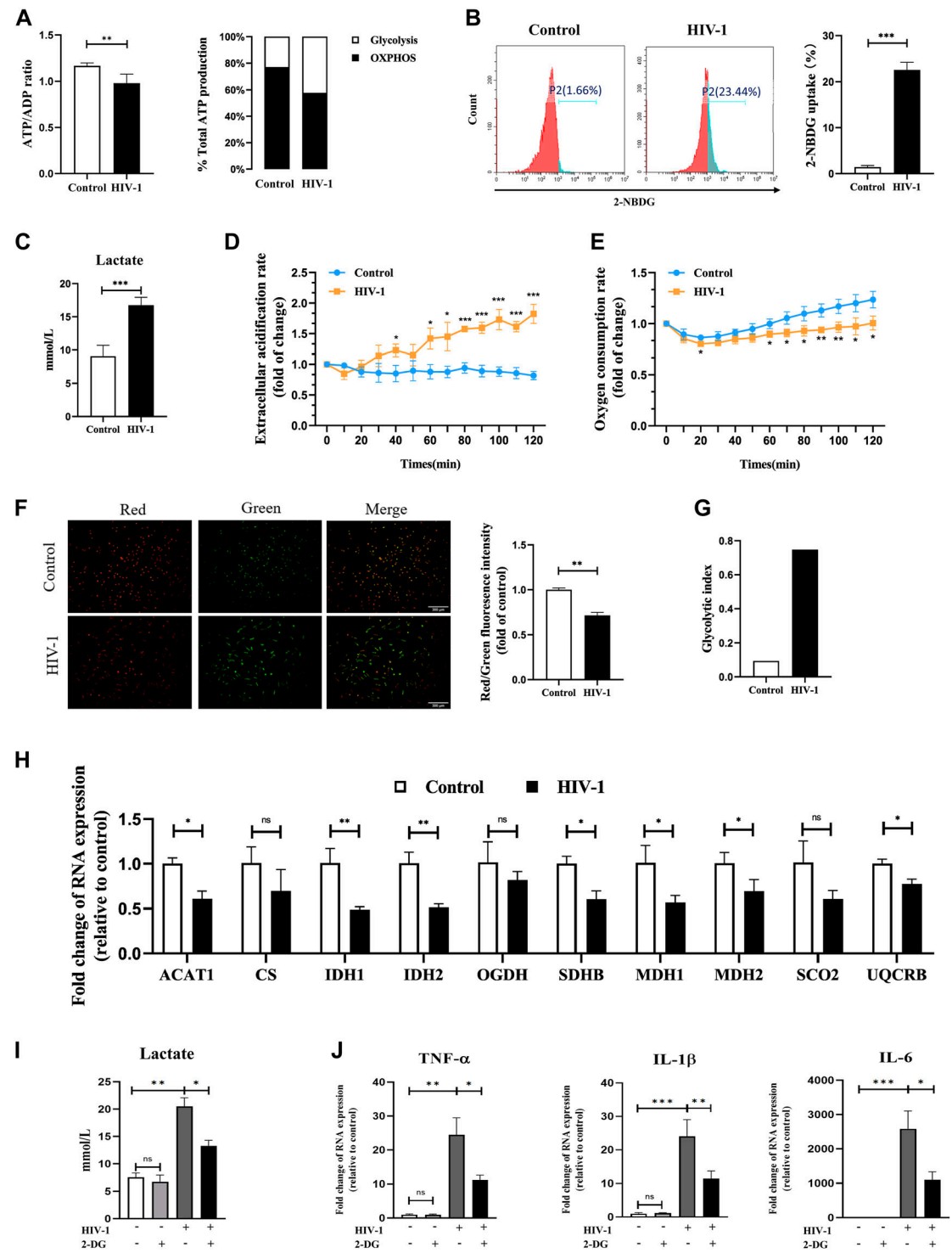

**Figure 2. Metabolism reprogramming affects polarization in HIV-1–infected macrophages.**
**(A)** ATP/ADP ratio (left panel) and relative ATP production from glycolysis and oxidative phosphorylation (right panel) in monocyte-derived macrophages (MDMs) infected or uninfected with HIV-1. (Statistical analysis was performed using a *t* test, *P < 0.05, **P < 0.01, and ***P < 0.001.) **(B)** Representative results of glucose uptake analyzed by flow cytometry. (Statistical analysis was performed using a *t* test, ***P < 0.001.) **(C)** Lactate concentration in the supernatant of MDMs infected with HIV-1 for 48 h was detected using colorimetry. (Statistical analysis was performed using a *t* test, ***P < 0.001.) **(D, E)** Extracellular acidification rate and oxygen consumption rate of MDMs infected with HIV-1 for 48 h. Time-resolved fluorescence was applied and monitored for 120 min. Results were relative to 0 min. (Statistical analysis was performed using a *t* test, *P < 0.05, **P < 0.01, and ***P < 0.001.) **(F)** Mitochondrial membrane potential was assessed using the JC-1 assay. A change from red to green fluorescence indicates a decrease in the mitochondrial membrane potential. The red/green fluorescence ratio was calculated for comparison. (Statistical analysis was performed using a *t* test, **P < 0.01.) Scale bar, 300 μm. **(G)** Bar plot showed glycolytic index (GI) differences in control and HIV-1–infected MDMs. **(H)** mRNA expressions of key genes in TCA and oxidative phosphorylation. (Statistical analysis was performed using a *t* test, ns non-significant, *P < 0.05, **P < 0.01, and ***P < 0.001.) **(I, J)** MDMs were pretreated

of HIF-1α, a known strong inducer of glycolysis, in HIV-1–infected MDMs. Immunofluorescence results showed that HIF-1α expression was enhanced after HIV-1 infection and was mainly located in the nucleus (Fig 3A). HIF-1α–activating genes, including HK2, PDK1, and LDHA (which function vitally in the glycolytic pathway), were also up-regulated by HIV-1 infection. However, in the presence of LW6 (15 μM), an inhibitor of HIF-1α, the HIV-1–induced up-regulation of HIF-1α, as well as its downstream targets HK2, PDK1, and LDHA, decreased to levels comparable to those in the control group (Fig 3B). The changes in expression of these factors at the protein level were also revalidated by Western blot, as shown in Fig 8B. As for key genes of OXPHOS, LW6 treatment had little effect on the down-regulated expression of TCA key genes induced by HIV-1 infection (Fig 3C). Meanwhile, treatment of another HIF-1α inhibitor YC-1 (1 μM) obtained similar results (Fig 3D and E). Next, we performed a lactate test and the JC-1 assay to determine the role of HIF-1α in HIV-1–induced glucose metabolic reprogramming. As shown in Fig 4A and B, elevated lactate induced by HIV-1 infection was effectively restrained under LW6 treatment, whereas the impaired mitochondrial activity was not restored, suggesting that HIF-1α activation contributes to glycolysis enhancement, but may not be a direct factor for the OXPHOS reduction in HIV-1 infection. In addition, we observed decreased expressions of TNF-α, IL-1β, and IL-6 after HIF-1α inhibition, both at mRNA (Fig 4C) and protein (Fig 4D) levels. Collectively, the above evidence suggests that HIV-1 infection activates the HIF-1α pathway to promote glycolysis and cause immunometabolic reprogramming in macrophages.

## HIV-1 regulates HIF-1α–dependent immunometabolic reprogramming via COX-2

Previous studies have demonstrated COX-2 activation motivates pro-inflammation, which is the main manifestation of M1 macrophages (Simon, 1999). In HIV-1–infected macrophages, we found COX-2 mRNA and protein expressions were markedly up-regulated (Fig 5A and B). Therefore, we sought to investigate whether the up-regulation of COX-2 is involved in mediating HIV-1–induced immunometabolic reprogramming in macrophages. Using meloxicam (50 μM), a selective inhibitor of COX-2 synthesis (Engelhardt, 1996), we found that meloxicam significantly reduced COX-2 expression induced by HIV-1 infection (Fig 5C). Key markers of glycolysis were subsequently detected. Results revealed that meloxicam treatment reversed the HIV-1–induced lactate production (Fig 5D), and also reversed the increased expression of HIF-1α and key enzymes in glycolysis (HK2, PDK1, and LDHA), resulting in a rate of glycolysis comparable to uninfected MDMs (Figs 5E and 8B). In addition, results of immunofluorescence exhibited obviously weakened fluorescence of HIF-1α after meloxicam treatment (Fig 5F and G), suggesting that COX-2 regulates the enhancement of glycolysis during HIV-1 infection through the HIF-1α pathway.

We subsequently correlated COX-2 activation with M1 polarization, showing that the expression of HIV-1–induced pro-inflammatory cytokines (TNF-α, IL-1β, and IL-6) was notably reversed by meloxicam treatment, particularly at the level of protein expression (Fig 5H and I).

To revalidate the role of COX-2 in regulating M1 polarization of macrophages, we constructed a COX-2–silenced THP-1 cell line using lentivirus-mediated RNAi (Fig 6A). Real-time quantitative PCR (RT-qPCR) and Western blot proved stable COX-2 knockdown expression in THP-1–derived macrophages (COX-2i group) (Fig 6B and C). In COX-2i cells, HIF-1α–dependent glycolysis and the expression of TNF-α, IL-1β, and IL-6 induced by HIV-1 were both reduced (Fig 6D–G), which was consistent with the results obtained from meloxicam treatment. These findings indicated that COX-2 plays an important role in mediating HIF-1α–dependent immunometabolic reprogramming of macrophages upon HIV-1 infection.

## Phosphorylation of JNK involves in HIV-1–regulating immunometabolic reprogramming in macrophages

To further explore the upstream events of COX-2 activation, we analyzed the differential genes between HIV-1–infected macrophages and control macrophages using bioinformatics techniques. Based on our transcriptome sequencing (RNA-seq) data, we found the MAPK signal transduction pathway was significantly enriched (Fig 7A). Gene set enrichment analyses revealed that genes involved in the MAPK pathway were more expressed in the HIV-1 infection group than those in the control group (Fig 7B, Table S1), supporting that MAPK signaling is associated with response to HIV-1 infection in macrophages. To further investigate which subpathway response is more sensitive to HIV-1 infection, we measured the expression of JNK, ERK1/2, and p38. At the mRNA level, we found that only JNK expression in the infection group was higher than that in the control group (P < 0.05), but there was no significant difference in the expression of total JNK at the protein level (Fig 7C–E). However, at the protein phosphorylation level, we found the ratio of p-JNK to total JNK protein was significantly higher in the HIV-1 infection group compared with the control group, whereas the ratios of p-ERK to ERK and of p-p38 to p38 had no significant differences between two groups (Fig 7C–E). Therefore, it is plausible that HIV-1 causes immune response of macrophages via activating the MAPK/JNK signaling pathway.

We then used SP600125, an inhibitor of JNK, to assess the role of JNK in HIV-1–induced immunometabolic reprogramming in macrophages. In the presence of SP600125 (50 μM), HIV-1–induced phosphorylation of JNK (p-JNK/total JNK) was reduced to the control level (uninfected with HIV-1 and untreated with SP600125), along with a dramatic reduction of HIV-1–induced COX-2 overexpression (Fig 8A), demonstrating the ability of JNK in regulating COX-2 expression. Meanwhile, under inhibition of JNK phosphorylation, the expression of key factors in glycolysis was significantly decreased (Fig 8B). In addition, to verify the role of COX-2 and HIF-1α in p-JNK–mediating metabolic reprogramming and M1 polarization, we also used meloxicam (50 μM), LW6 (15 μM), and YC-1 (4 μM) to specifically inhibit the COX-2 pathway and HIF-1α pathway, respectively. The results confirmed the inhibition of COX-2 or HIF-1α

with or without 2-DG (10 mM) for 2 h, followed by infection of HIV-1 for 48 h. **(I, J)** Lactate concentration in the culture supernatant (I), and mRNA expressions of TNF-α, IL-1β, and IL-6 (J) were detected. (Statistical analysis was performed using a t test, *P < 0.05, **P < 0.01, and ***P < 0.001).

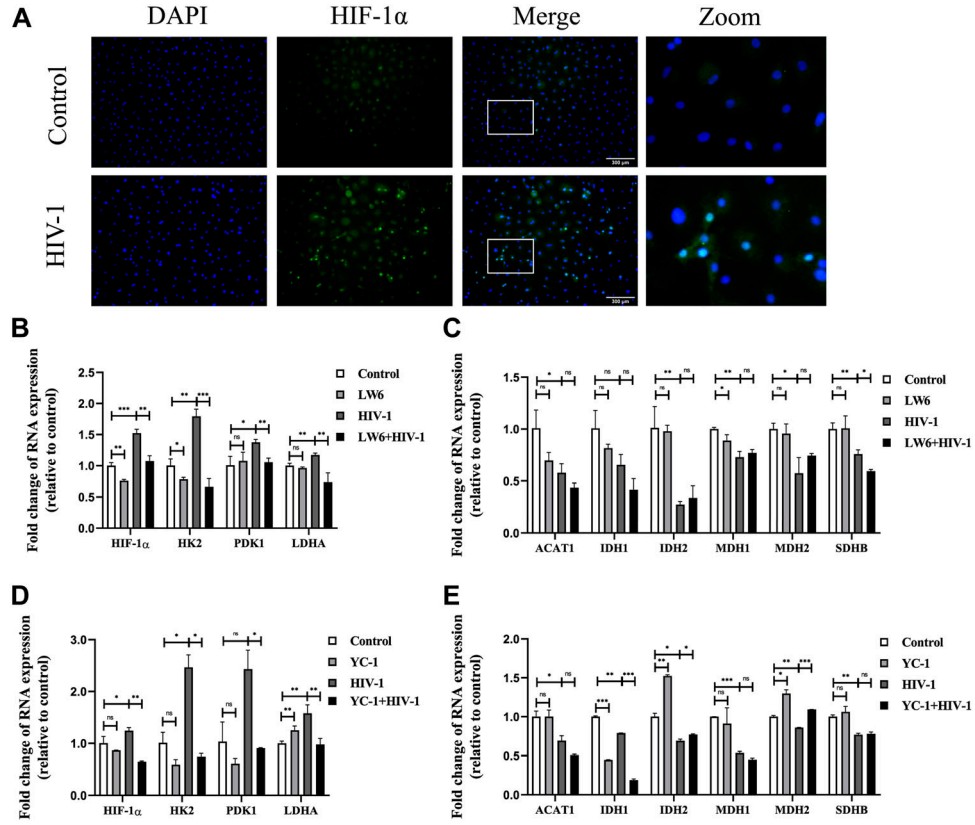

**Figure 3. Activation of HIF-1α impinges upon gene regulation of glucose metabolism.**
**(A)** Immunofluorescence showed HIF-1α protein localization and expression in monocyte-derived macrophages (MDMs). Blue: DAPI; green: HIF-1α. Scale bar, 300 $\mu$m. **(B, C)** MDMs were pretreated with or without LW6 (15 $\mu$M) for 2 h, followed by HIV-1 infection for another 48 h. **(B, C)** RT–qPCR analysis showed the mRNA levels of key genes in the glycolysis (B) and TCA cycle (C). (Statistical analysis was performed using a $t$ test, *$P$ < 0.05, **$P$ < 0.01, and ***$P$ < 0.001.) **(D, E)** MDMs treated with or without YC-1 (1 $\mu$M) and infected with or without HIV-1 were assessed for expressions of key genes in the glycolysis (D) and TCA cycle (E). (Statistical analysis was performed using a $t$ test, *$P$ < 0.05, **$P$ < 0.01, and ***$P$ < 0.001).

led to the decreased expression of key factors (HIF-1α, HK1, HK2, and LDHA) in glycolysis (Fig 8B), which provided evidence at the protein level to support previous results in Figs 3 and 5. As a result of p-JNK inhibition, the HIV-1–induced lactate displayed a 30–50% reduction compared with HIV-1 infection alone, indicating the weakened glycolysis (Fig 8C). Furthermore, the inhibition of p-JNK, along with the suppression of COX-2 and HIF-1α, as well as the inhibition of glycolysis, resulted in decreased M1 polarization. Most notably, expressions of TNF-α and IL-6 were not increased under SP600125 treatment, despite the strong M1 induction from HIV-1 demonstrated above (Fig 8D and E).

Taken together, by analyzing changes in COX-2 expression, glycolysis level, and M1 polarization in the presence or absence of JNK inhibition, we found that phosphorylation of JNK promotes glycolysis enhancing, which is regulated by the COX-2/HIF-1α signaling axis, supporting M1 polarization of macrophages in HIV-1 infection.

## Discussion

In this study, using in vitro models of HIV-1–infected MDMs and THP-1 macrophages, we observed that macrophages were mainly driven to M1-like changes, with a large amount of pro-inflammatory cytokines produced, whereas the expression of M2-related markers was slightly changed. This is largely consistent with previous reports in vivo or in vitro (Porcheray et al, 2006; Cassol et al, 2010;

Burrack & Morrison, 2014), showing that macrophages tend to be M1-polarized during HIV-1 infection. The current understanding of M1 polarization in macrophages is mainly based on studies of LPS and IFN-γ stimulation. It has been found that JAK/STAT (Runtsch et al, 2022), PI3K/Akt (Vergadi et al, 2017), or Notch (Lin et al, 2018) signaling pathways are involved in polarization of macrophages under different stimuli or different microenvironments (Yunna et al, 2020). However, the molecular mechanism by which HIV-1 regulates macrophage polarization is not fully understood.

Immune cells adopt diverse metabolic strategies for different activation states, which are essential to resist pathogen invasion and maintain microenvironmental homeostasis (Russell et al, 2019). For example, activated T cells promote glycolysis and production of proteins, as well as nucleic acids, whereas Tregs and memory T cells rely on oxidization of fatty acids for fuel (Maciolek et al, 2014). For macrophages, evidence suggests that IFN-γ– or other pro-inflammatory factor–induced M1 macrophages use the glycolysis pathway and the pentose phosphate pathway as energy sources, whereas TCA, OXPHOS, and FAO are down-regulated. In contrast, M2 macrophages increase FAO, glutaminolysis, and mitochondrial respiration (Saha et al, 2017; Viola et al, 2019). In the context of HIV infection, Palmer and co-workers have revealed that CD4[+] T cells activated by HIV-1 infection switch metabolic phenotype from oxidative metabolism to aerobic glycolysis (Palmer et al, 2014). In addition, another study showed that central memory HIV-1–specific CD8[+] T cells have a metabolic profile characterized by elevated glycolysis with activation of the mTORC1 pathway (Angin et al, 2019).

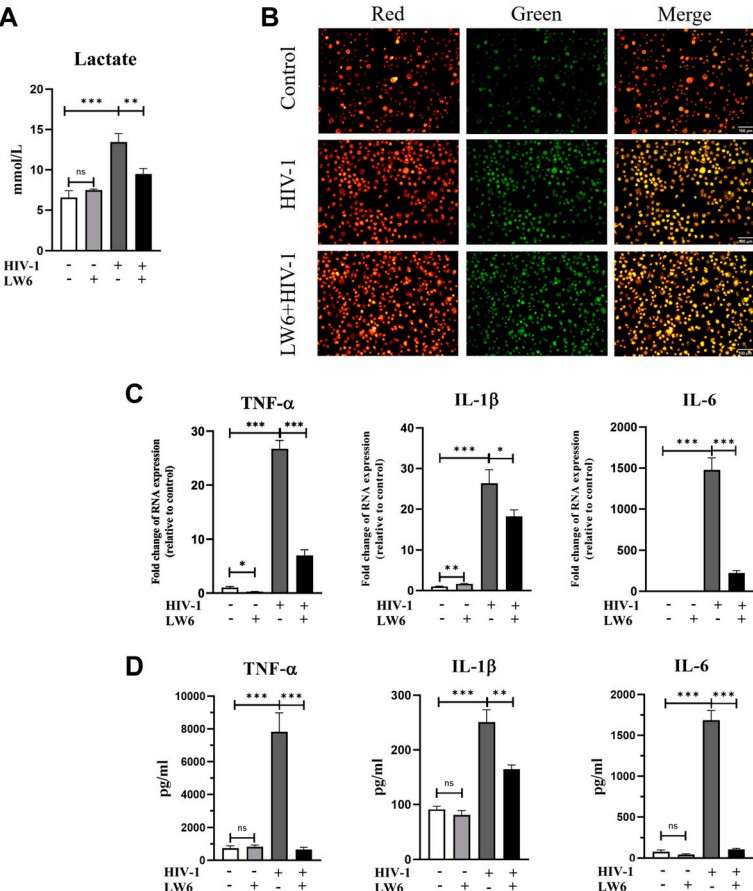

**Figure 4. Effect of HIF-1α inhibition on glycolysis, mitochondrial activity, and M1 polarization in monocyte-derived macrophages (MDMs).**
MDMs were infected with HIV-1 for 48 h, in the presence or absence of LW6 (15 $\mu$M) pretreatment (2 h). **(A)** Lactate concentration in the culture supernatant of MDMs. (Statistical analysis was performed using a *t* test, **$P < 0.01$ and ***$P < 0.001$.) **(B)** Mitochondrial membrane potential was assessed by the JC-1 assay. (Statistical analysis was performed using a *t* test, *$P < 0.05$.) Scale bar, 100 $\mu$m. **(C, D)** mRNA (C) and protein levels (D) of TNF-α, IL-1β, and IL-6 were detected using RT–qPCR and ELISA, respectively. (Statistical analysis was performed using a *t* test, *$P < 0.05$, **$P < 0.01$, and ***$P < 0.001$).

On the contrary, it has been shown that OXPHOS of primary myeloid dendritic cells (mDCs) from elite controllers (ECs) with very low levels of HIV-1 infection is increased compared with both HIV-1–positive individuals undergoing combination antiretroviral therapy and healthy individuals (Hartana et al, 2021). In this study, we turn our attention to the changes in glucose metabolism in macrophages upon HIV-1 infection. By measuring glucose uptake, lactate concentration, OCR, and ECR, we found enhanced glycolysis and impaired TCA and OXPHOS in HIV-1–infected macrophages, which are similar to those observed in T cells by Palmer (Palmer et al, 2014). However, there are some articles that take different views. For example, Castellano et al found that HIV-1 infection of macrophages leads to lipid accumulation and TCA regulation disorder, but does not cause changes in glycolysis (Castellano et al, 2019). This discrepancy may be due to the different cell culture protocols. The usage of M-CSF in the culture of macrophages in Castellano's study could promote M2 polarization (Chen et al, 2021), which is accompanied by a different metabolic profile from that of M1 macrophages (Na et al, 2015). To avoid the influence of M-CSF on macrophage metabolism, in our study, the primary macrophages were differentiated without additional cytokine stimulation, as some previous studies described (Sharma et al, 2022).

Macrophage M1 polarization, accompanied by the release of a large number of pro-inflammatory cytokines, is an important source of chronic inflammation in HIV-1–infected patients.

Currently, the use of glucose to locate inflamed tissues has been clinically applied, such as [18]F-fluorodeoxyglucose–positron emission tomography ([18]F-FDG-PET) imaging (Rudd et al, 2002; Sarrazin et al, 2012). Hammoud et al performed [18]F-FDG-PET in HIV patients and found those with high levels of glycolysis in CD4[+] T cells and monocytes before ART initiation were more likely to develop immune reconstitution inflammatory syndrome (Hammoud et al, 2019). This suggests that increased glycolysis may be related to the inflammatory status in HIV-1–infected patients. In the present study, by blocking glycolysis with 2-DG or heptelidic acid, we observed reversal of the increased pro-inflammatory cytokines induced by HIV-1 infection, indicating that macrophages prime glycolytic shift to facilitate M1 polarization in HIV-1 infection, further pointing to the clinical potential of reducing inflammation in HIV-1–infected patients by correcting glucose metabolism. It is worth noting that glucose metabolic regulation of immune cells can also influence HIV-1 pathogenesis. The study of CD4[+] T cells has shown that HIV-1 infection is strongly impaired when glucose metabolism is inhibited (Valle-Casuso et al, 2019). Consistent with that, in this study, HIV-1–infected macrophages showed a significant increase in ATP production, particularly from glycolysis. After inhibiting glycolysis, we found that HIV-1 replication was reduced (Fig S3). Enhanced glucose metabolism favors HIV-1 infection, perhaps because metabolically active cells have increased the susceptibility to HIV-1 infection. Another possibility is that metabolic activation produces molecules necessary for HIV-1 survival (Sáez-Cirión et al, 2021).

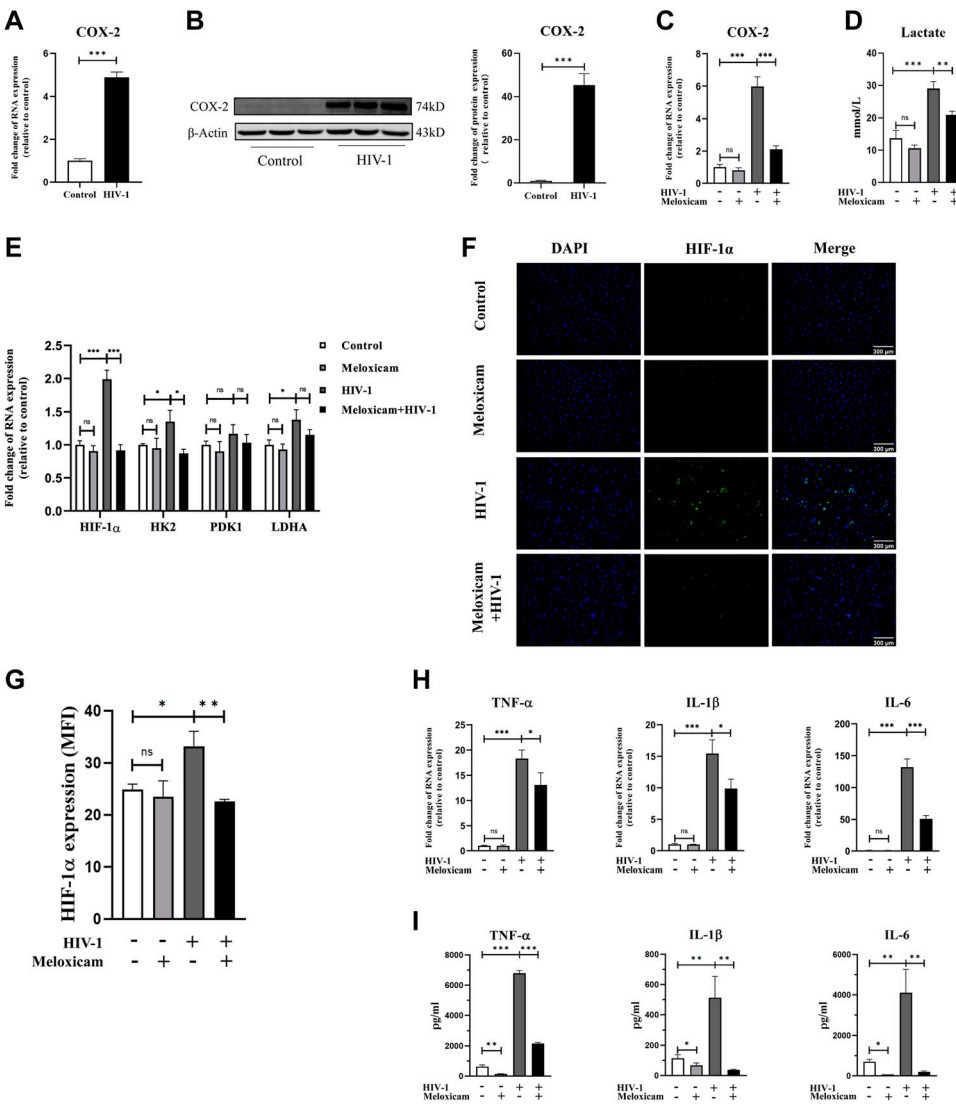

**Figure 5.** **Activation of COX-2 mediates HIF-1α–dependent glucose metabolism reprogramming and M1 polarization in monocyte-derived macrophages (MDMs).** **(A, B)** mRNA (A) and protein expressions (B) of COX-2 in MDMs after HIV-1 infection for 48 h. β-Actin was used as the normalization control in Western blot analysis. (Statistical analysis was performed using a t test, **P < 0.001.) **(C, D, E, F, G, H, I)** MDMs were pretreated with meloxicam (50 μM) for 2 h; then, HIV-1 particles were added to incubate for another 48 h. COX-2 expression (C), lactate production (D), key glycolytic gene expression (E), hypoxia-inducible factor 1α expression ((F), scale bar, 300 μm), and M1 polarization (H, I) were measured. **(G)** Quantification of hypoxia-inducible factor 1α fluorescence intensity. (Statistical analysis was performed using a t test, *P < 0.05, **P < 0.01, and ***P < 0.001).

In the present study, we found that HIF-1α was markedly up-regulated by HIV-1 infection, and inhibition of HIF-1α reduced HIV-1–induced glycolysis and M1 polarization. Activation of glycolysis by HIF-1α is considered critical for immunometabolic adaptation in monocytes (Cheng et al, 2014; Kim et al, 2006). As a transcription factor, HIF-1α mediates the expression of glycolytic enzymes (Semenza et al, 1994). Previous studies have revealed that HK is up-regulated in HIV-1–infected CD4[+] T cells (Kavanagh Williamson et al, 2018). Consistently, we observed glycolytic enzymes such as HK1/2, PDK1, and LDHA were up-regulated in HIV-1–infected macrophages, whereas inhibition of HIF-1α repressed the expression of glycolytic enzymes. The evidence suggests that HIV-1–manipulated immunometabolism may be achieved through the regulation of multiple glycolytic enzymes, and HIF-1α serves as the pivotal mediator in this progress. Because HIF-1α activation has also been reported to suppress mitochondrial function in cancer (Semenza, 2007; Nagao et al, 2019), we speculate that the deceased mitochondrial respiration in HIV-1–infected macrophages may be regulated by HIF-1α. We then tried two inhibitors of

HIF-1α; however, neither of them recovered the attenuated OXPHOS. In fact, in peritoneal macrophages derived from Hif-1α [-/-] mice, the level of OXPHOS was also not elevated (Yu et al, 2020). Our results, combined with previous studies, suggest that the regulatory mechanism of OXPHOS in macrophages is specific to different tumor microenvironments or pathogen invasions. Therefore, HIV-1–mediated mitochondrial damage in macrophages was HIF-1α–independent, and other mechanisms are worthy of further investigation.

We further explored the upstream events of HIV-1–induced HIF-1α expression and found that COX-2 was significantly up-regulated in macrophages when exposed to HIV-1. COX-2 is rarely expressed in steady-state cells, whereas it is boosted in activated cells (O'Banion, 1999). Traditionally, COX-2 mediates inflammatory response through PGE2 production (Mizuno et al, 2019). Interestingly, recent studies have found that in addition to PGE2, COX-2 can also regulate the HIF-1α/PKM2 pathway in apoptosis resistance (Wang et al, 2019). In HIV-1–induced M1 polarization, we observed that the activation of HIF-1α, enhancement of glycolysis, and pro-

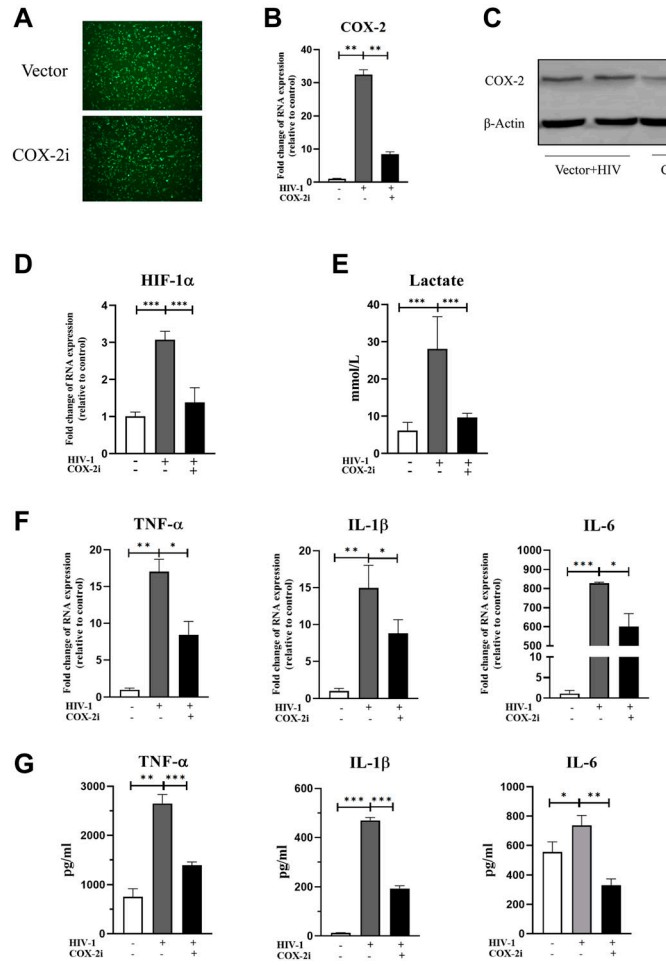

**Figure 6. Knockdown of COX-2 represses HIF-1α–dependent glycolytic shift and M1 polarization in THP-1 macrophages.**
**(A)** EGFP fluorescence of THP-1 macrophages transfected with lentivirus. **(B, C)** COX-2–silencing THP-1 macrophages were verified by RT–qPCR and Western blot. (Statistical analysis was performed using a t test, **P < 0.01.) **(D, E, F, G)** HIF-1α expression (D), lactate production (E), and mRNA and protein expressions of TNF-α, IL-1β, and IL-6 **(F, G)** were decreased in COX-2–silencing THP-1 macrophages, as quantified by RT–qPCR and ELISA. (Statistical analysis was performed using a t test, *P < 0.05, **P < 0.01, and ***P < 0.001).

inflammatory cytokine secretion were effectively reversed under COX-2 inhibition, indicating that HIV-1 promotes HIF-1α–dependent glycolysis and M1 polarization through COX-2.

As one of the central pathways in various cellular programs, the MAPK signaling pathway has been demonstrated to participate in immune regulation through COX-2 (Cargnello & Roux, 2011; Lin et al, 2015). Here, we investigated the role of phosphorylation of JNK, ERK, or p38 in regulation of COX-2 by HIV-1, and found that HIV-1 promoted phosphorylation of JNK, but not ERK and p38. In addition, we confirmed that COX-2 was positively regulated by JNK phosphorylation in the context of HIV-1 infection. Currently, the role of the MAPK signaling pathway in glucose metabolism remains controversial. For instance, MAPKs increase glucose uptake during exercise, whereas they mediate glucose intolerance in metabolic syndrome (Schultze et al, 2012; Bengal et al, 2020). In this study, inhibition of JNK suppressed HIF-1α–dependent glycolysis and reduced secretion of pro-inflammatory factors, supporting that HIV-1–induced immunometabolic reprogramming in macrophages is modulated by JNK and its downstream COX-2 activation.

In summary, the present study suggests that M1 polarization of macrophages is related to glycolysis promotion via the JNK/COX-2/HIF-1α signaling axis in HIV-1 infection. The findings broaden our understanding of mechanisms underlying macrophage polarization, and imply that alleviating glycolysis may be a strategy to reduce inflammatory levels in individuals living with HIV-1.

# Materials and Methods

## Ethics statement

Guangxi Medical University Ethical Committee approved all experimental procedures and protocols used in this study (Approval No. 2022-0130).

## Reagents and antibodies

PMA, LPS, IFN-γ, IL-4, IL-10, and SP600125 were purchased from Sigma-Aldrich. 2-Deoxy-D-glucose (2-DG), heptelidic acid (HA), oligomycin (Omy), LW6, and YC-1 were from MCE (United States). Meloxicam was from TCI (Japan). Immunofluorescence blocking buffer (#12411), and Alexa Fluor 488 goat anti-rabbit (#4412; RRID: AB_1904025) and Alexa Fluor 594 goat anti-mouse (#8890; RRID: AB_2714182) secondary antibodies were purchased from Cell Signaling Technology. Antibodies against β-actin (#3700; RRID: AB_2242334), phospho-SAPK/

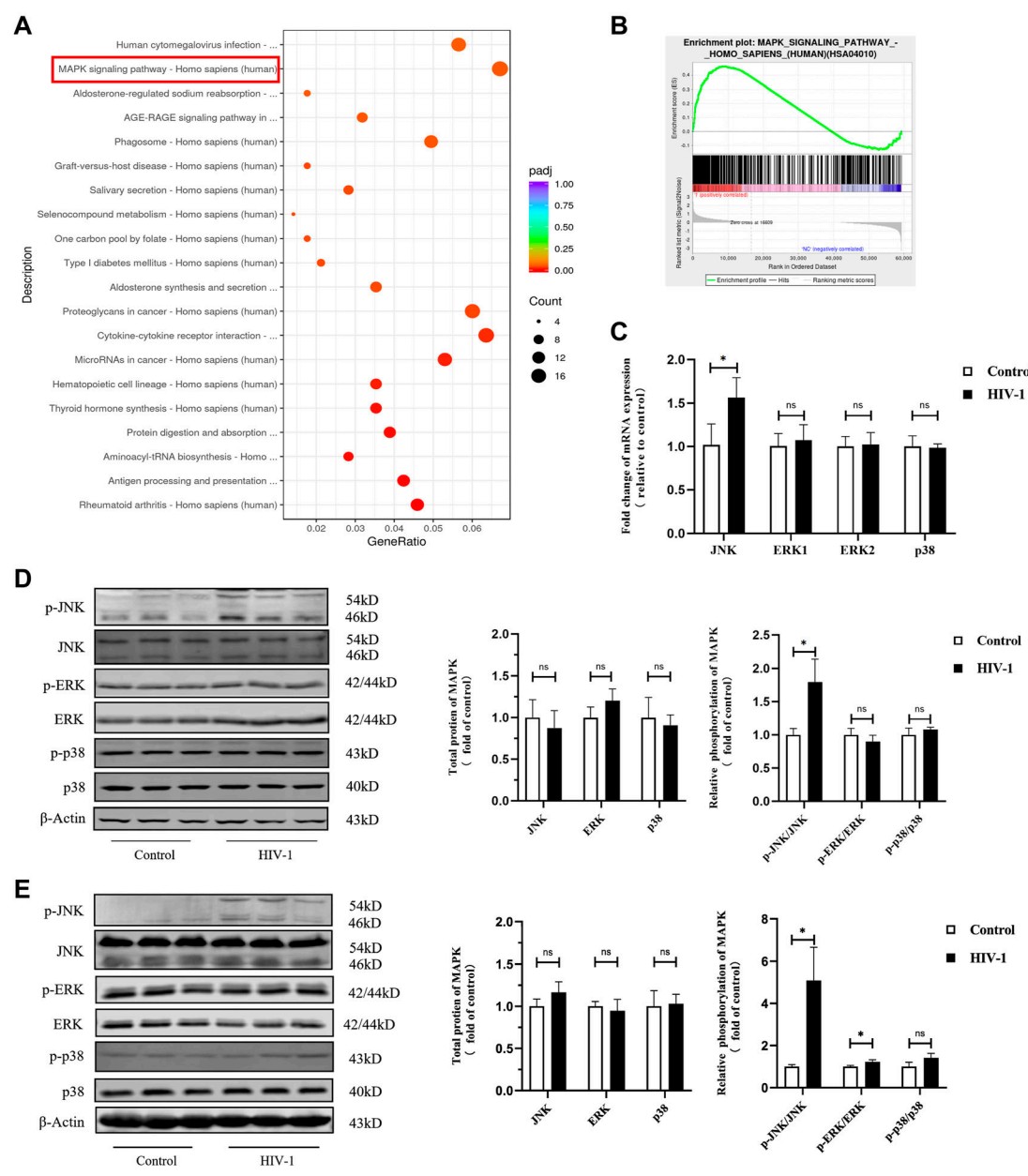

**Figure 7. HIV-1 urges phosphorylation of JNK in macrophages.**
THP-1 macrophages and primary monocyte-derived macrophages (MDMs) were infected with or without HIV-1 for 48 h. **(A)** KEGG enrichment analysis of differentially expressed genes in HIV-1–infected THP-1 macrophages and control macrophages. The size of the bubble positively correlated with the number of enriched genes. The x-axis represents the gene ratio, and the color of the bubble represents the adjusted *P*-value of enrichment analysis. **(B)** Gene set enrichment analysis was performed in Control and HIV-1 groups. In this figure, the y-axis represents enrichment score (ES), and on the x-axis are genes (vertical black lines) included in gene sets. The analysis demonstrates that the MAPK signaling pathway is enriched in the HIV-1 group (NES = 1.315, *P*-value < 0.0001, FDR = 0.301). The detailed information is provided in Table S1. **(C)** RT–qPCR analysis showed the mRNA levels of JNK, ERK, and p38 in MDMs. (Statistical analysis was performed using a *t* test, *$P$ < 0.05.) **(D, E)** Western blot confirmed phosphorylation of JNK is increased in MDMs (D) and THP-1 macrophages (E) upon HIV-1 infection. (Statistical analysis was performed using a *t* test, *$P$ < 0.05).

JNK (#4668; RRID: AB_823588), SAPK/JNK (#9252; RRID: AB_2250373), phospho-p44/42 MAPK (Erk1/2) (#4370; RRID: AB_2315112), p44/42 MAPK (Erk1/2) (#4695; RRID: AB_390779), phospho-p38 MAPK (#4511; RRID: AB_2139682), p38 MAPK (#8690; RRID: AB_10999090), COX-2 (#12282; RRID: AB_2571729), and HIF-1α (#36169; RRID: AB_2799095) and Glycolysis Antibody Sampler Kit (#8337; RRID: AB_10897509) were obtained from Cell Signaling Technology. The secondary antibodies

of IRDye 680RD donkey anti-mouse IgG and IRDye 800CW donkey anti-rabbit IgG were purchased from LI-COR.

## Cell culture and differentiation

The human monocyte THP-1 cell line (conserved in our laboratory, ATCC number: TIB-202; RRID: CVCL_0006) and primary human

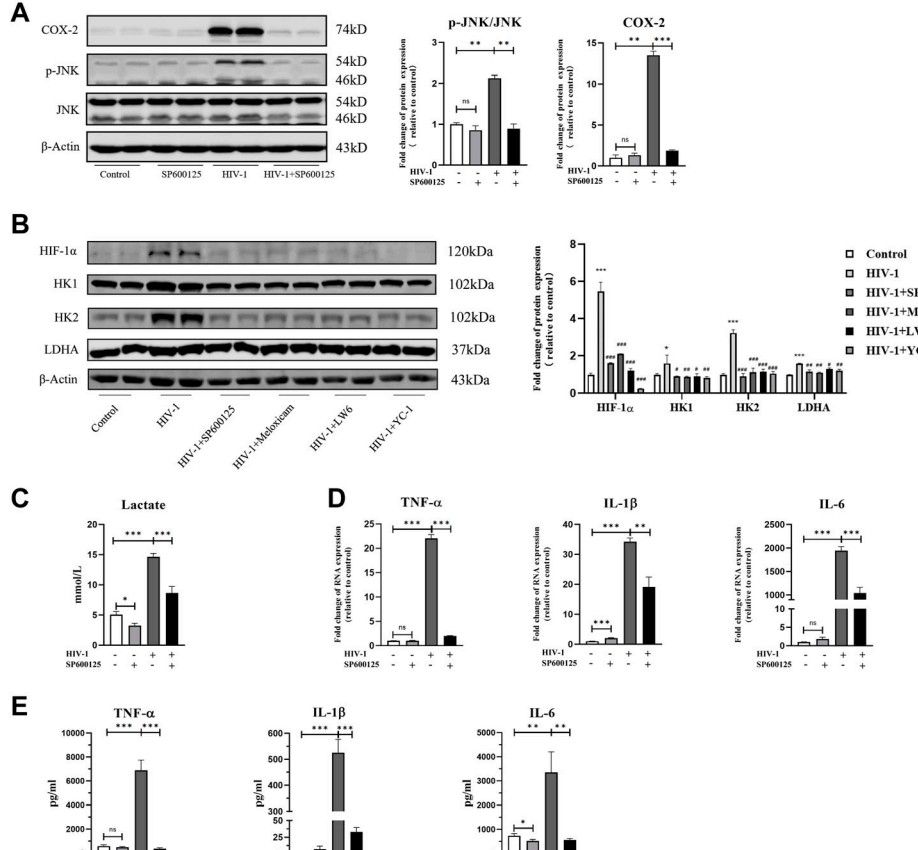

**Figure 8. SP600125 reverses HIV-1–induced M1 polarization by inhibiting JNK activation.**
Monocyte-derived macrophages (MDMs) were pretreated with or without SP600125 (50 $\mu$M) for 2 h, followed by infection of HIV-1 for 48 h. **(A)** Expressions of JNK and COX-2 were measured by Western blot. Total JNK was used as the normalization control of phosphorylated JNK, and $\beta$-actin was used as the normalization control of COX-2. (Statistical analysis was performed using a $t$ test, *$P < 0.05$, **$P < 0.01$, and ***$P < 0.001$.) **(B)** Western blot analysis of hypoxia-inducible factor 1$\alpha$, HK1, HK2, and LDHA in MDMs. $\beta$-Actin was used as the normalization control. SP600125, meloxicam, LW6, and YC-1 were administered as mentioned above. (Statistical analysis was performed using a $t$ test, *$P < 0.05$ and **$P < 0.01$ as compared to Control; #$P < 0.05$, ##$P < 0.01$, and ###$P < 0.001$ as compared to HIV-1.) **(C)** Lactate concentration in the culture supernatant of MDMs was detected using colorimetry. (Statistical analysis was performed using a $t$ test, *$P < 0.05$ and ***$P < 0.001$.) **(D, E)** MDMs were collected, and the expression of TNF-$\alpha$, IL-1$\beta$, and IL-6 was measured by RT–qPCR (D) and ELISA (E). (Statistical analysis was performed using a $t$ test, *$P < 0.05$, **$P < 0.01$, and ***$P < 0.001$).

macrophages were used as cell models for HIV-1 infection. THP-1 cells were routinely cultured in 1640 medium (Gibco) containing 10% heat-inactivated FBS (Gibco) and 1% penicillin–streptomycin solution (Solarbio) at 37°C and 5% $CO_2$ atmosphere. To obtain THP-1–derived macrophages, THP-1 cells were seeded in a six-well culture plate ($10^6$ cells per well) with DMEM (Gibco) containing 10% FBS and 1% penicillin–streptomycin solution, and stimulated with 50 ng/ml of PMA (48 h) for differentiation (Dunn et al, 2011). Primary human macrophages were derived from PBMCs of healthy donors as previously described (Fang et al, 2009). Briefly, PBMCs were isolated from whole blood by Ficoll (GE Healthcare) through density gradient centrifugation according to the manufacturer's protocol (Riedhammer et al, 2016). PBMCs were then incubated in 75-cm$^2$ flasks precoated with gelatin (VETEC) in a 37°C, 5% $CO_2$ atmosphere for 30 min. Adhesive monocytes were harvested using EDTA (Solarbio). Collected monocytes were seeded at a density of $4 \times 10^6$ cells/ml in complete 1,640 medium (containing 10% human A$^+$B$^+$ serum) and refreshed medium every 3 d until differentiation into macrophages for experiments.

## Virus production and infection

HIV-1$_{Bal}$ virions were originally from the NIH AIDS Reagent Program and reproduced in our laboratory. The modified method was based on Edward's report (Barker et al, 1998). In brief, PHA (5 $\mu$g/ml) in complete 1,640 medium (containing 100 U/ml IL-2) stimulated human PBMCs for 3 d at 37°C, 5% $CO_2$. Afterward, polybrene (2 $\mu$g/ml) was added to enhance HIV-1 infection. After HIV-1$_{Bal}$ infection, the supernatant was harvested every 3 d and viral titers were quantified using a HIV-1 p24 ELISA kit (Jianglaibio). We adjusted the viral titer to 100 ng p24/ml and stored it at –80°C.

HIV-1 infection was performed after differentiation of monocytes into macrophages. HIV-1 virions were added to culture medium (8 ng p24/$10^6$ cells) for 6 h. Then, cells were washed with PBS (Solarbio) and cultured in fresh medium until harvest at the appointed time.

## Lentiviral construction and transduction

Lentiviral particles were obtained from GeneChem. In detail, the siRNAs for COX-2 (NM_000963.4) (COX-2i) were chimeric into the GV248 plasmid. The sequence for the Vector was as follows: 5′-TTCTCCGAACGTGTCACGT-3′. The sequence for COX-2i was as follows: 5′-GCTGAATTTAACACCCTCTAT-3′. Titers of concentrated viral particles were $8.0 \times 10^8$ units/ml. Lentiviral particles were transduced into THP-1 cells during the logarithmic growth phase. The supernatant was removed after 24 h of infection and replaced with complete 1,640 medium. To generate stable COX-2 knockdown cell lines, 2 $\mu$g/ml puromycin was added to medium for 48 h. A fluorescent microscope was used to observe successful transduction.

RT-qPCR and Western blot were used to confirm the down-regulated expression of COX-2 in COX-2i THP-1 cells compared with Vector THP-1 cells.

## RNA extraction and RT-qPCR

Total cellular RNA was extracted using TRIzol (Sigma-Aldrich). The cells were pipetted with TRIzol (40 $\mu$l/$10^5$ cells) until completely lysed, and chloroform (Tedia) was added for stratification by the gradient. After centrifugation (4°C, 12,000 g/min, 15 min), the transparent supernatant was collected. Isopropanol (Tedia) was used to precipitate RNA, and 75% ethanol was for purification. The remaining RNA pellet was resuspended in enzyme-free water, followed by incubation at 55°C for 15 min. The quantification was performed using NanoDrop 2000C (United States). The cDNA was synthesized with PrimeScript RT Master Mix (Takara) using 1 $\mu$g RNA. cDNA samples were used for RT-qPCR with SYBR Premix Ex Taq™ (Takara) according to the operation manual. The sequences of primer pairs are listed in Table S2. Relative mRNA expressions were analyzed based on the $2^{-\Delta\Delta Ct}$ method, normalizing to the house-keeping gene GAPDH.

## Cytokine and lactate measurements

Culture supernatants were collected at the indicated time points, and protein concentrations of TNF-$\alpha$, IL-1$\beta$, and IL-6 were quantified by ELISA according to the manufacturer's instructions (NOVUS). Lactate production was measured by a lactate kit (Njjcbio) using colorimetry according to the manufacturer's instructions.

## Flow cytometry

For detection of polarization markers of macrophages, MDMs were harvested by scraping. The harvested cells were washed with PBS to remove the culture medium thoroughly and collected by centrifugation. For intracellular staining, the cells were fixated and permeabilized using the Cytofix/Cytoperm kit (BD Biosciences), then stained with PE-linked anti-CD68 (#12-0689-41, RRID: AB_10805746; eBioscience), APC-linked anti-CD86 (#17-0869-41, RRID: AB_2802218; eBioscience), and APC-linked anti-CD206 (# 17-2069-42, RRID: AB_2573182; eBioscience) for 20 min at 4°C. The stained cells were washed with PBS and finally resuspended in PBS. Flow cytometry was performed using a CytoFLEX2 flow cytometer (Beckman Coulter) and analyzed by FlowJo software.

## Western blot

Protein samples were prepared using lysis buffer (RIPA: PMSF = 100:1, RIPA: phosphatase inhibitors = 100:1). The total protein concentration of samples was quantified using the BCA Protein Assay kit (Beyotime) to make sure each sample had an equal amount of protein for SDS–PAGE. After transferring to PVDF membranes, the membranes were blocked with 5% skim milk for 1 h at RT and then incubated with primary antibody overnight at 4°C. Fluorescent secondary antibody was used for detection. Relative band intensities were measured by the LI-COR Odyssey CLx imaging system.

## ATP assay and ATP/ADP ratio assay

ATP concentrations were measured using CellTiter-Glo Kit (Promega) following the manufacturer's protocol. Differentiated macrophages were infected with or without HIV-1 for 48 h in refreshed medium, and incubated with detection reagents at 37°C for 10 min. The fluorescence intensity was read under a microplate reader (Synergy H1; BioTek). ATP concentrations were determined and normalized to the protein level of each sample.

Changes in the ATP/ADP ratio were detected using ATP/ADP-Lite Assay Kit (Vigorous Bio). The treated MDMs were lysed with lysis buffer for 10 min at RT with gentle shaking. Subsequently, 50 $\mu$l of assay reagent was mixed with 10 $\mu$l of the cell lysate, and the samples were then analyzed using a microplate reader to obtain the initial readings (Data A). After a 10-min incubation, another reading was taken (Data B). To each sample, 10 $\mu$l of ATP converting enzyme was added, followed by another reading after 10 min (Data C). The ATP/ADP ratio was calculated using the following formula:

$$ATP/\ ADP\ ratio = Data\ A/(Data\ C - Data\ B)$$

## Glucose uptake assay

Glucose uptake was measured using 2-NBDG Glucose Uptake Assay Kit (BioVision) according to the manufacturer's instructions. Briefly, MDMs ($10^5$ cells) were seeded in a 24-well plate with 400 $\mu$l detection reagent (376 $\mu$l 0.5% serum 1,640 cell culture medium, 4 $\mu$l 2-NBDG reagent, and 20 $\mu$l glucose uptake enhancer) at 37°C for 30 min. Subsequently, cells were washed with iced analysis buffer three times and scraped off with rubber policeman. Fluorescence intensity was examined by a Beckman flow cytometer (excitation at 488 nm).

## Extracellular oxygen consumption and acidification test

Changes in oxygen consumption and acidification after HIV-1 infection were analyzed using assays from Abcam (#ab197243 and #ab197244). Macrophages were seeded in a 96-well plate (black wall) at a density of $10^5$ cells/well. The cell culture medium was replaced with an extracellular oxygen consumption reagent or acidification assay reagent, and fluorescence was monitored using the time-resolved fluorescence (Synergy H1; BioTek). Results were normalized to the total DNA content, which was assessed with Hoechst dye (R&D).

## MMP detection

MMP was measured by JC-1 (Beyotime). MDMs were seeded in a 48-well plate. JC-1 dye was prepared according to the instructions and added to the plate for incubation at 37°C for 20 min. The cells were then washed with iced washing buffer. Images were acquired using a fluorescence microscope, and image processing was performed using ImageJ software.

### Immunofluorescence staining

MDMs were prepared for immunofluorescence staining. 4% form-aldehyde and 100% cold methanol were used for fixation and permeation, respectively. Blocking was performed with blocking buffer at RT for 60 min. Cells were then incubated with rabbit anti-HIF-1α (1:500) or anti-β-actin (1:5,000) overnight at 4°C. The immune complexes were glowed using Alexa Fluor 488 goat anti-rabbit (1:1,000) or Alexa Fluor 594 goat anti-mouse (1:1,000) secondary antibodies before counterstaining with DAPI (Solarbio). Images were obtained using a fluorescence microscope (EVOS FL Auto2; Invitrogen), and image processing was performed using ImageJ software.

### Statistical analysis

All experiments were repeated at least three times. Statistical analysis was performed using a $t$ test with SPSS v25.0 and GraphPad Prism v8.0. Data were presented as the mean ± SD, and $P < 0.05$ was considered as the significance of differences between the two groups.

# Data Availability

All data are available in the article or the supplementary materials. Raw data generated during the current study will be available from the corresponding author on reasonable request.

# Supplementary Information

# Acknowledgements

This work was supported by the Guangxi Youth Science Foundation Project (2021JJB140599), National Natural Science Foundation of China (31970167, 31860040, 81971935), Middle-aged and Young Teachers' Basic Ability Promotion Project of Guangxi (2021KY0088), and Guangxi Science and Technology Planning Project (AD21220013).

## Author Contributions

J Zhang: investigation, methodology, and writing—original draft.
Z Yuan: conceptualization, funding acquisition, and project administration.
X Li: methodology.
F Wang: formal analysis.
X Wei: software.
Y Kang: validation.
C Mo: validation.
J Jiang: conceptualization and funding acquisition.
H Liang: supervision and writing—review and editing.
L Ye: data curation, funding acquisition, and writing—review and editing.

## Conflict of Interest Statement

The authors declare that they have no conflict of interest.

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
