## [Reviewer comments · Life Science Alliance]

Life Science Alliance

Activation of JNK/COX-2/HIF-1 α axis promotes M1 macrophage via glycolytic shift in HIV-1 infection

Junhan Zhang, Zongxiang Yuan, Xuanrong Li, Fengyi Wang, Xueqin Wei, Yiwen Kang, Chuye Mo, Junjun Jiang, Hao Liang and Li Ye

DOI: <https://doi.org/10.26508/lsa.202302148>

Corresponding author(s): Prof. Li Ye (Guangxi Medical University) and Hao Liang (Guangxi Medical University)

Review Timeline:

Submission Date:	2023-05-10
Editorial Decision:	2023-06-19
Revision Received:	2023-09-15
Editorial Decision:	2023-09-20
Revision Received:	2023-09-26
Accepted:	2023-09-27

Transaction Report:

June 19, 2023

Re: Life Science Alliance manuscript #LSA-2023-02148-T

Li Ye
Guangxi Medical University
22nd Shuangyong Road
Nanning 530021
CHINA

Dear Dr. Ye,

Thank you for submitting your manuscript entitled "Activation of JNK/COX-2/HIF-1 α axis promotes M1 macrophage via glycolytic shift in HIV-1 infection" to Life Science Alliance. The manuscript was assessed by expert reviewers, whose comments are appended to this letter. We invite you to submit a revised manuscript addressing the Reviewer comments.

Thank you for this interesting contribution to Life Science Alliance. We are looking forward to receiving your revised manuscript.

Sincerely,

B. MANUSCRIPT ORGANIZATION AND FORMATTING:

Reviewer #1 (Comments to the Authors (Required)):

Authors report on metabolic reprogramming of macrophages upon HIV-1 infection.

Although not all data are completely novel (M1 reprogramming upon HIV infection was shown before and should be mentioned in the introduction), the characterization of the "Warburg effect" in macrophages upon HIV infection is new and well characterized in this work.

Comments

1. replace outdated terminology HAART with cART (combination antiretroviral therapy)

2; the effect of the inhibitor of glycolysis (2DG) is small. please explain.

3. typo's on p6

clarify HIV-1 23 44% and in control 1 66% ???

1 7 times, sloppy, is it 1.7 or 17?

4. for discussion: Is anything known about metabolic alteration upon HIV infection in other target cells?

Reviewer #2 (Comments to the Authors (Required)):

In this study titled 'Activation of JNK/COX-2/HIF-1 α axis promotes M1 macrophage via glycolytic shift in HIV-1 infection' Zhang et al investigated whether HIV-1 infection can drive pro-inflammatory activation of human macrophages. HIV-1 infection was found to increase the expression and release of pro-inflammatory cytokines in human monocyte-derived macrophages and THP1-derived macrophages. HIV-1 infection was shown to increase in aerobic glycolysis in a HIF-1 dependent manner. Inhibition of glycolysis with 2-DG or HIF-1 inhibition reduced HIV-1-induced pro-inflammatory cytokine production. Furthermore, HIV-1 infection was shown to increase JNK phosphorylation and COX-2 expression, and this is suggested to stabilise HIF-1 to drive glycolytic reprogramming and cytokine expression. The conclusions of the authors are primarily derived from pharmacological approaches with potential caveats and would need further validation with other experimental approaches.

Major comments:

1. The authors need to better establish the requirement of glycolysis for pro-inflammatory cytokine production following HIV-1 infection. 2-DG has previously been shown to reduce glycolysis and oxidative phosphorylation in macrophages (PMID: 30184486). The GAPDH inhibitor Heptelidic/Koningic acid and genetic silencing of hexokinase could be employed as an orthogonal approach to 2-DG. HIV-1 infection of primary human macrophages should also be performed under glucose-deprived and galactose-substituted conditions with measurements rates of glycolysis (lactate release and ECAR), mitochondrial respiration, and pro-inflammatory cytokines.

2. The ATP/ADP ratio should be measured, as opposed to ATP concentrations. The impact of OXPHOS inhibition using oligomycin on HIV-1-mediated macrophage activation should be compared to approaches targeting glycolysis.

3. The role of HIF-1 is drawn from two pharmacological approaches LW6 and YC-1. However, the blot in figure 8(b) shows that there is still a significant level of HIF-1 α following LW6 treatment. As LW6 is thought to promote HIF-1 α degradation this needs to be addressed before any conclusions can be drawn from these results. Additionally, there is no western blot of HIF-1 α with YC-1 treatment, only RNA levels are presented in figure 3(d).

4. Meloxicam is a non-steroidal anti-inflammatory drug (NSAIDs), some of which have been reported to activate the anti-inflammatory transcription factor NRF2, in addition to inhibiting COX activity (PMID: 35588739). Therefore, it is unclear if the impact of meloxicam on HIF-1, glycolysis or cytokines can be attributed to COX-2. It is unclear how COX-2 would regulate HIF-

1 stability. COX-2 silencing was only shown to decrease cytokines but it's impact of HIF-1 stabilization, target gene expression or glycolysis (lactate release and ECAR) was not demonstrated.

Minor comments:

1. The addition of hypoxic score (Figure 2h) does not add value to this paper and it's not clear what genes this score is comprised.
2. The quantification of HIF-1 α nuclear localisation is not given (Figure 3a).
3. TMRM is a preferred method of measuring changes in mitochondrial membrane potential whereas JC-1 is best used in cases where mitochondrial membrane potential is altered drastically e.g. cell death
4. There is no description of microscopy settings, or the microscope used.

Dear Prof. Guidi

Many thanks for the opportunity to revise our manuscript “**Activation of JNK/COX-2/HIF-1 α axis promotes M1 macrophage via glycolytic shift in HIV-1 infection**” (Manuscript ID LSA-2023-02148-T). We have carefully addressed all comments and provided our point-by-point responses as follows:

Reviewer1

Comment 1 replace outdated terminology HAART with cART (combination antiretroviral therapy)

Response: We thank the reviewer for pointing this out. We have made the necessary updates accordingly.

Comment 2 the effect of the inhibitor of glycolysis (2DG) is small. please explain.

Response: Thank you for your careful review. The effect of 2-DG depends on the concentration used, in our study, we chose a relatively low concentration (10 mM) of 2-DG treatment, which showed that 10mM 2-DG had a significant inhibitory effect on lactate release and inflammatory cytokines, but did not completely block them ($P < 0.05$, Fig 2j-k). We did not choose higher concentrations of 2-DG for treatment because 2-DG has cytotoxic effect, and the toxicity varies in different cells. For example, at 10 mM treatment, only 20% of MDA-MB-231 cells survived (DOI: 10.1007/s10549-018-4874-z IF: 4.87 Q2), while 90% of BXPC-3 cells survived (DOI: 10.1016/j.biopha.2019.109521 IF: 7.5 Q1). Based on the evaluation of the toxicity of 2-DG in macrophages by CCK8 assessment, we chose a concentration of 10 mM for follow-up experiments, because 2-DG at 20 mM has a significant cytotoxic effect on macrophages (as shown in the figure below). In addition, as suggested by reviewer 2, we added the treatment of GAPDH inhibitor Heptelidic acid to further confirm the impact of glycolysis inhibition on M1 polarization. We found that M1 polarization was inhibited under Heptelidic acid treatment, which is consistent with the results of 2-DG treatment (Fig S3), indicating the results are valid.

[Figure removed by editorial staff per authors' request]

Comment 3 typo's on p6

Response: We appreciate your comment and have corrected the error.

Comment 4 for discussion: Is anything known about metabolic alteration upon HIV infection in other target cells?

Response: We thank the reviewer for this good suggestion. As you suggested, we have now included a statement in the Discussion section as follows: “In the context of HIV infection, Palmer

and co-workers have revealed that CD4⁺ T cells activated by HIV-1 infection switch metabolic phenotype from oxidative metabolism to aerobic glycolysis (Palmer et al, 2014). Additionally, another study showed that central memory HIV-1-specific CD8⁺ T cells have a metabolic profile characterized by elevated glycolysis with activation of mTORC1 pathway (Angin et al, 2019). On the contrary, it has been shown that oxidative phosphorylation of primary myeloid dendritic cells (mDCs) from elite controllers (ECs) with very low levels of HIV-1 infection is increased compared to both HIV-1-positive individuals undergoing cART and healthy individuals (Hartana et al, 2021).”

Reviewer2

Major comments

1. The authors need to better establish the requirement of glycolysis for pro-inflammatory cytokine production following HIV-1 infection. 2-DG has previously been shown to reduce glycolysis and oxidative phosphorylation in macrophages (PMID: 30184486 [IF: 31.373 Q1]). The GAPDH inhibitor Heptelidic/Koningic acid and genetic silencing of hexokinase could be employed as an orthogonal approach to 2-DG. HIV-1 infection of primary human macrophages should also be performed under glucose-deprived and galactose-substituted conditions with measurements rates of glycolysis (lactate release and ECAR), mitochondrial respiration, and pro-inflammatory cytokines.

Response: We highly appreciate your comments. As you suggested, we added experiments under Heptelidic acid treatment, and the results showed that Heptelidic acid treatment inhibited M1 polarization (Fig S3), which is consistent with the results of 2-DG treatment. These results confirm the requirement of glycolysis for pro-inflammatory cytokine production following HIV-1 infection.

We are grateful for the suggestion to measure the rates of glycolysis, mitochondrial respiration, and pro-inflammatory cytokines under glucose-deprived and galactose-substituted conditions. However, considering that HIV-1 infection and replication are severely impaired in the absence of glucose (DOI: 10.1038/s41577-020-0381-7 IF: 100.3 Q1), and that various metabolic pathways, including oxidative phosphorylation, are greatly affected (DOI: 10.1158/0008-5472.can-03-1101 IF: 11.2 Q1; DOI:10.1371/journal.pone.0070772 IF: 3.7 Q2), we don't think glucose-deprived and galactose-substituted experiments are necessary. The present study focused on the effect of HIV-1 infection rather than glucose on macrophage metabolism. Nevertheless, we agree that determining the effect of glucose on the reprogramming of cellular immune metabolism and subsequently on HIV infection and replication would be an interesting issue to explore.

2. The ATP/ADP ratio should be measured, as opposed to ATP concentrations. The impact of OXPHOS inhibition using oligomycin on HIV-1-mediated macrophage activation should be compared to approaches targeting glycolysis.

Response: We appreciate your comment. In the revised manuscript, we replaced ATP concentrations with ATP/ADP ratios (Fig 2a) and examined the effect of Oligomycin on HIV-1-mediated macrophage activation (Fig S3).

3. The role of HIF-1 α is drawn from two pharmacological approaches LW6 and YC-1. However, the blot in figure 8(b) shows that there is still a significant level of HIF-1 α following LW6 treatment. As LW6 is thought to promote HIF-1 α degradation, this needs to be addressed before any conclusions can be drawn from these results. Additionally, there is no western blot of HIF-1 α with YC-1 treatment, only RNA levels are presented in figure 3(d).

Response: We thank the reviewer for this question. Under hypoxia condition, LW6 may not completely degrade HIF-1 α , and the degradation degree varies with the concentration and treatment time (DOI: 10.1016/j.bcp.2010.06.018 IF: 5.8 Q1). In our original manuscript, the degradation of HIF-1 α by LW6 treatment was not obvious in Western Blot (WB), which may be related to the concentration used. To confirm our conjecture, we performed a dose response experiment and found that 15 μ M LW6 significantly promoted HIF-1 α degradation in macrophages (as shown in the figure below). We also conducted a dose-response experiment of YC-1 and found that HIF-1 α was significantly degraded by 1 μ M, 2 μ M, 4 μ M YC-1 treatment for 24 hours (as shown in the figure below). In the revised manuscript, we have replaced the Figure 8b with these new results.

[Figure removed by editorial staff per authors' request]

4. Meloxicam is a non-steroidal anti-inflammatory drug (NSAIDs), some of which have been reported to activate the anti-inflammatory transcription factor NRF2, in addition to inhibiting COX activity (PMID: 35588739 IF: 43.474 Q1). Therefore, it is unclear if the impact of meloxicam on HIF-1 α , glycolysis or cytokines can be attributed to COX-2. It is unclear how COX-2 would regulate HIF-1 α stability. COX-2 silencing was only shown to decrease cytokines but its impact of HIF-1 α stabilization, target gene expression or glycolysis (lactate release and ECAR) was not demonstrated.

Response: This is a good question. To address the issue of Meloxicam's effect on NRF2 expression in macrophages, we examined the expression of NRF2 in the presence or absence of meloxicam. The results showed that meloxicam had no significant effect on NRF2 expression, while stably inhibited COX-2 expression (MG-132 was used as an inducer of NRF2). Therefore, in macrophages, meloxicam affects HIF-1 α , glycolysis or cytokines by inhibiting COX-2 rather than activating NRF2. Regarding how COX-2 regulates HIF-1 α stability, in this study, we use meloxicam and lentivirus knockdown to inhibit COX-2, and both showed reduced HIF-1 α expression (Fig 5f, Fig 6d). It has been reported that COX-2 interacts with HIF-1 α through co-localization in nuclear (DOI: 10.1007/s10549-010-1240-1 IF: 3.8 Q2). Thus, it is possible that COX-2 maintains HIF-1 α stability through direct binding. However, due to the susceptibility of HIF-1 α to degradation under normoxic conditions (as evidenced in our previous experiment where HIF-1 α induced by HIV-1 was undetectable after 24 hours of cell lysis), CO-IP experiments involving HIF-1 α in other studies typically necessitate the construction of cell lines with HIF-1 α overexpression and are conducted within a hypoxia workstation (DOI:10.1016/j.bbamcr.2015.01.011 IF: 3.51; Q1). Therefore, it is difficult for us to conduct CO-IP experiments in the absence of both conditions. Alternatively, we evaluated the

binding of COX-2 (PDB ID: 5F19) and HIF-1 α (PDB ID: 4H6J) by molecular docking with HDock software (DOI: 10.1038/s41596-020-0312-x IF: 13.49; Q1). Fig b below shows the binding model of two proteins, and the docking score and confidence score in Table 1 below indicate a high probability of binding between the two proteins (docking score < -200 and confidence score > 0.7 represent a high probability of binding). Taken the above information together, COX-2 may increase the stability of HIF-1 α by binding with HIF-1 α . Moreover, we will continue to investigate the interaction of COX-2 and HIF-1 α in the future once experimental conditions permit. As you suggested, we have added the impact of COX-2 silencing on HIF-1 α expression and glycolysis in revised manuscript (Fig 6d,e).

[Figure and table removed by editorial staff per authors' request]

Minor comments

1. The addition of hypoxic score (Figure 2h) does not add value to this paper and it's not clear what genes this score is comprised.

Response: We have removed Figure 2h and the corresponding description.

2. The quantification of HIF-1 α nuclear localisation is not given (Figure 3a).

Response: Thank you. The quantitative results are shown in Fig 5g, and the average fluorescence intensity was obtained by three independent experiments.

3. TMRM is a preferred method of measuring changes in mitochondrial membrane potential whereas JC-1 is best used in cases where mitochondrial membrane potential is altered drastically e.g. cell death

Response: We appreciate the reviewer for this comment. JC-1 is also a reliable and sensitive probe for the detection of mitochondrial membrane potential, and has been successfully used in various studies to measure MMP changes in different cell types and under different stimuli, such as oxidative stress (doi: 10.1016/j.redox.2018.04.008 IF: 11.4 Q1), hypoxia (DOI: 10.3389/fnmol.2023.1216947 IF: 5.64; Q1) and drug treatment (doi: 10.1186/1476-4598-10-95 IF: 37.3 Q1), not limited to drastic cytological changes. Compare to TMRM, JC-1 has the capability of dual-emission that provides internal control for potential-independent effects and minimizes artifacts caused by changes in dye concentration and loading time. TMRM, on the other hand, is a

single-emission dye that needs to be corrected with another Mito Tracker or Hoechst staining. Therefore, we believe that the JC-1 results are sufficient to demonstrate mitochondrial damage in HIV-infected macrophages.

4. There is no description of microscopy settings, or the microscope used.

Response: We have added the microscopy information in the Methods Section as follows: “Images were obtained using a fluorescence microscope (EVOS FL Auto2, Invitrogen, United States), and image processing was performed using Image J software.” For the microscopy settings, a scale bar was marked on the picture to indicate the magnification of the view.

We would like to thank the Editor and the Reviewers again for the valuable comments and suggestions on our manuscript. We have addressed these comments and revised the manuscript extensively. We hope that our revised manuscript will meet with your approval for publication.

Sincerely Yours,

Li Ye
Guangxi Medical University

September 20, 2023

RE: Life Science Alliance Manuscript #LSA-2023-02148-TR

Prof. Li Ye
Guangxi Medical University
22nd Shuangyong Road
Nanning, Nanning 530021
China

Dear Dr. Ye,

Thank you for submitting your revised manuscript entitled "Activation of JNK/COX-2/HIF-1 α axis promotes M1 macrophage via glycolytic shift in HIV-1 infection". We would be happy to publish your paper in Life Science Alliance pending final revisions necessary to meet our formatting guidelines.

- please add ORCID ID for the corresponding (and secondary corresponding) author--you should have received instructions on how to do so
- please add the Twitter handle of your host institute/organization as well as your own or/and one of the authors in our system
- please remove the Graphical Abstract from the manuscript file and leave it uploaded separately with the file designation "Graphical Abstract"
- we encourage you to revise the figure legends for figures 6 and 8 such that the figure panels are introduced in alphabetical order
- the activities listed for authors Li Ye and Junjun Jiang in the submission system and in the Author Contributions section do not qualify them for authorship. please either update or let us know if an author should be removed.

A. FINAL FILES:

B. MANUSCRIPT ORGANIZATION AND FORMATTING:

Sincerely,

Reviewer #2 (Comments to the Authors (Required)):

The authors have addressed most of my comments sufficiently and have provided further supporting evidence. The manuscript is suitable for acceptance in it's current form.

September 27, 2023

RE: Life Science Alliance Manuscript #LSA-2023-02148-TRR

Prof. Li Ye
Guangxi Medical University
22nd Shuangyong Road
Nanning, Nanning 530021
China

Dear Dr. Ye,

Thank you for submitting your Research Article entitled "Activation of JNK/COX-2/HIF-1 α axis promotes M1 macrophage via glycolytic shift in HIV-1 infection". It is a pleasure to let you know that your manuscript is now accepted for publication in Life Science Alliance. Congratulations on this interesting work.

DISTRIBUTION OF MATERIALS:

Again, congratulations on a very nice paper. I hope you found the review process to be constructive and are pleased with how the manuscript was handled editorially. We look forward to future exciting submissions from your lab.

Sincerely,
